# Synergistic Photodynamic/Antibiotic Therapy with Photosensitive MOF-Based Nanoparticles to Eradicate Bacterial Biofilms

**DOI:** 10.3390/pharmaceutics15071826

**Published:** 2023-06-26

**Authors:** Lehan Du, Wenjun Shi, Xin Hao, Liang Luan, Shibo Wang, Jiaju Lu, Quan Zhang

**Affiliations:** 1Institute of Smart Biomedical Materials, School of Materials Science and Engineering, Zhejiang Sci-Tech University, Hangzhou 310018, China; 2Department of Laboratory Medical Center, General Hospital of Northern Theater Command, No. 83, Wenhua Road, Shenhe District, Shenyang 110016, China

**Keywords:** antibacterial agents, biofilms, metal–organic frameworks, nanoparticles, photodynamic therapy

## Abstract

Bacterial biofilms pose a serious threat to human health, as they prevent the penetration of antimicrobial agents. Developing nanocarriers that can simultaneously permeate biofilms and deliver antibacterial agents is an attractive means of treating bacterial biofilm infections. Herein, photosensitive metal–organic framework (MOF) nanoparticles were developed to promote the penetration of antibiotics into biofilms, thereby achieving the goal of eradicating bacterial biofilms through synergistic photodynamic and antibiotic therapy. First, a ligand containing benzoselenadiazole was synthesized and incorporated into MOF skeletons to construct benzoselenadiazole-doped MOFs (Se-MOFs). The growth of the Se-MOFs could be regulated to obtain nanoparticles (Se-NPs) in the presence of benzoic acid. The singlet oxygen (^1^O_2_) generation efficiencies of the Se-MOFs and Se-NPs were evaluated. The results show that the Se-NPs exhibited a higher ^1^O_2_ generation efficacy than the Se-MOF under visible-light irradiation because the small size of the Se-NPs was conducive to the diffusion of ^1^O_2_. Afterward, an antibiotic drug, polymyxin B (PMB), was conjugated onto the surface of the Se-NPs via amidation to yield PMB-modified Se-NPs (PMB-Se-NPs). PMB-Se-NPs exhibit a synergistic antibacterial effect by specifically targeting the lipopolysaccharides present in the outer membranes of Gram-negative bacteria through surface-modified PMB. Benefiting from the synergistic therapeutic effects of antibiotic and photodynamic therapy, PMB-Se-NPs can efficiently eradicate bacterial biofilms at relatively low antibiotic doses and light intensities, providing a promising nanocomposite for combating biofilm infections.

## 1. Introduction

Bacterial antibiotic resistance has emerged as a significant global threat to public health [1,2]. Diseases caused by drug-resistant bacterial infection have become more difficult to treat, thus promoting and increasing the serious spread of these diseases and the risk of death [3,4]. Approximately 80% of human chronic microbial infections are due to bacterial biofilms on living tissues [5]. Biofilms are intricate communities of microorganisms consisting of one or more bacterial species that are embedded within a self-produced matrix of extracellular polymeric substances. The primary constituents of this matrix are polysaccharides, proteins, nucleic acids, and lipids that are secreted by the resident bacteria [6,7]. The dense extracellular matrix in the biofilm prevents the diffusion of antibiotics and the host’s immune cells into the interior, leading to bacterial resistance to various antimicrobial agents by the special self-protection mechanism [8]. Biofilms are very difficult to remove completely, thus leading to repeated infections, treatment failure, and even death [9,10]. Therefore, the development of alternative and effective strategies for eradicating bacterial biofilms is of paramount importance [11,12].

Photodynamic therapy (PDT) has garnered significant attention as an effective method to kill bacteria and eradicate bacterial biofilms [13]. In PDT, a photosensitizer (PS) is utilized to selectively absorb light of a specific wavelength and subsequently transfer energy to molecular oxygen, resulting in the production of reactive oxygen species (ROSs). The generation of ROSs by PSs occurs through two distinct mechanisms: the type I mechanism, resulting in radical species such as hydroxyl radicals and superoxide anions, and the type II mechanism, resulting in the formation of singlet oxygen (^1^O_2_) [14,15]. ROSs can react with biological molecules and cause photo-oxidation of bacterial cell membranes and DNA, leading to bacterial necrosis and apoptosis [16,17]. Many PSs, including porphyrin [18,19], methylene blue [20,21], phthalocyanine [22], and BODIPY dyes [23], have been developed for antibacterial PDT. However, the photodynamic efficiencies of these PSs in eliminating bacterial biofilms are always not high [24,25]. One of the main reasons is the limited penetration of PSs, which is severely hindered by dense extracellular polymers. As a result, only a small number of PSs can reach and interact with bacteria inside biofilms, leading to reduced treatment efficiency [26].

Metal–organic frameworks (MOFs) are a class of compounds formed by the coordination of metal ions or clusters with organic ligands [27,28]. Due to their tunable compositions and porous structures, MOFs have been widely studied as drug delivery carriers for antibacterial treatments [29]. Various antibiotics [30,31], nucleic acids [32], enzymes [33,34], and proteins [35] have been investigated so far as cargo for drug delivery. MOFs possess porous structures that enable the encapsulation of diverse drugs and therapeutic agents. These active components can be immobilized within the cavities of the MOFs via covalent or non-covalent interactions [36,37]. In addition, MOFs can serve as carriers for PS delivery [38,39]. In recent years, researchers have designed and synthesized different kinds of PS ligands to prepare photosensitive MOFs [40,41]. Incorporating PSs into MOF skeletons can avoid PS aggregation and self-quenching in physiological environments, facilitating the diffusion of ^1^O_2_ and high efficiency in PDT [42]. Despite great progress in the photodynamic inactivation of bacteria using MOFs, there is still an urgent need to develop photosensitive MOFs capable of effectively eradicating bacterial biofilms at low antibiotic doses and light intensities [43,44].

Recently, we described a facile approach for synthesizing photosensitive MOFs with mixed ligands for the photodynamic inactivation of bacteria [45]. To confer photodynamic properties upon MOFs, a benzoselendiazole-containing ligand (Se-TPDC) was incorporated into MOF skeletons, forming benzoselendiazole-doped MOFs (Se-MOFs). Although Se-MOFs have been demonstrated to effectively inhibit the formation of bacterial biofilms, their size precludes their passage through the intricate mesh of biofilm matrices and the narrow diffusion channels between bacterial clusters, thereby restricting their ability to penetrate and diffuse within a biofilm. Additionally, the poor diffusion and limited penetration of Se-MOFs within biofilms result in therapeutic failure. Thus, the development of nanoscale MOFs might be an effective way to enhance their biofilm permeability. In this study, we used benzoic acid as a regulator to control the growth of MOFs and obtained photosensitive MOF-based nanoparticles (Se-NPs), as shown in Figure 1A. The ^1^O_2_ generation efficiency of the Se-NPs was compared with that of the Se-MOFs under visible-light irradiation. To further improve the efficiency of the biofilm eradication, an antibiotic drug, polymyxin B (PMB), was coupled onto the surface of the Se-NPs via amidation to yield PMB-modified Se-NPs (PMB-Se-NPs). The antibacterial mechanism of the PMB-Se-NPs was systematically studied using flow cytometry. Finally, the photodynamic efficiency of the PMB-Se-NPs in eradicating bacterial biofilms was assessed using a biofilm model of *Pseudomonas aeruginosa* bacteria (Figure 1B).

## 2. Materials and Methods

### 2.1. Preparation of Se-MOF

Me-TPDC and Se-TPDC were synthesized using the previously described method [45]. A Se-MOF was synthesized by dissolving Me-TPDC (50 mg, 0.14 mmol) and Se-TPDC (15.2 mg, 0.035 mmol) in 100 mL of N,N-dimethylformamide (DMF), followed by the addition of anhydrous ZrCl_4_ (42 mg, 0.18 mmol) and acetic acid (1.08 mL, 0.018 mmol) to the solution. The addition of anhydrous ZrCl_4_ as a metal source promoted the formation of the Se-MOF, while acetic acid was utilized as a regulator during the synthesis process. The resulting mixture was stirred at 105 °C for 48 h to yield the desired product. The prolonged reaction time at elevated temperatures facilitated the formation of strong and stable bonds between the various components of the mixture. After the completion of the reaction, the resulting mixture was cooled to room temperature and the crude product was collected with centrifugation. The collected solid powder was then washed thrice with DMF and methanol, respectively. Finally, the product was dried for 48 h under vacuum at an ambient temperature to obtain a Se-MOF.

### 2.2. Preparation of Se-NPs

Se-NPs were synthesized with a solvothermal method. Me-TPDC (50 mg, 0.14 mmol) and Se-TPDC (15.2 mg, 0.035 mmol) were first dissolved in DMF (36 mL), followed by the addition of anhydrous ZrCl_4_ (42 mg 0.18 mmol). To further regulate the size and morphology of the Se-NPs, benzoic acid (22 mg, 0.18 mmol) and Tween 80 (840 mg) were added to the aforementioned mixture solution. The benzoic acid was used as a coordination competitor to regulate the growth and aggregation of the Se-NPs during synthesis. Meanwhile, the Tween 80 acted as a surfactant to promote the dispersion and stability of the Se-NPs in the solution. After 5 min of sonication, the mixture was transferred to a 100 mL polytetrafluoroethylene (PTFE) hydrothermal synthesis reactor. The PTFE reactor was maintained at 105 °C for 24 h. The resulting mixture was centrifuged at room temperature to obtain Se-NPs, which were subsequently washed thrice with DMF and methanol, respectively. To ensure complete desiccation, the Se-NPs were dried under vacuum for 48 h at an ambient temperature.

### 2.3. Synthesis of PMB-Se-NPs

PMB-Se-NPs were synthesized through the covalent attachment of PMB onto the surfaces of Se-NPs. A mixture of Se-NPs (40 mg), 1-(3-dimethylaminopropyl)-3-ethylcarbodiimide hydrochloride (EDC · HCl) (306.7 mg, 1.6 mmol), and N-hydroxysuccinimide (NHS) (184.1 mg, 1.6 mmol) in phosphate-buffered saline (PBS, 24 mL) was stirred for 2 h in an oil bath at 35 °C. During this period, the carboxyl groups on the surfaces of the Se-NPs were activated to form stable active esters. Then, a solution of PMB in PBS (10 mg mL^−1^, 1 mL) was added to the mixture and stirred for 24 h. PMB-Se-NPs were obtained with centrifugation and then washed thrice with water. 

Quantification of the PMB content was carried out using the classical ninhydrin hydrate method [46]. Briefly, ninhydrin hydrate (100 mg, 0.56 mmol) and stannous chloride dihydrate (48 mg, 0.2 mmol) were dissolved in 50 mL of hot water to prepare a 0.2% hydrated ninhydrin stock solution. After the reaction supernatant (2 mL) was diluted, an acetic acid buffer (1 mL, pH 6) and a ninhydrin solution (1 mL) were added and incubated in boiling water for 15 min. The absorbance value at 570 nm was detected using a microplate reader (Varioskan LUX, ThermoFisher SCIENTIFIC, Waltham, MA, USA). The concentration of PMB was calculated based on a standard curve established using a series of PMB solutions with known concentrations.

### 2.4. Intracellular ROS Assay

Intracellular ROS generation was assessed via flow cytometry using 2′,7′-dichlorofluorescein diacetate (DCFH-DA) as a fluorescent probe. Monoclonal colonies of Pseudomonas aeruginosa (*P. aeruginosa*, ATCC 9027) were incubated in Luria–Bertani (LB) liquid medium (6 mL) for 12 h under shaking conditions (160 rpm, 37 °C). Subsequently, the bacterial suspension was standardized to a concentration of 1 × 10^7^ CFU mL^−1^ by adjusting the optical density at a wavelength of 600 nm (OD_600nm_ = 0.1, corresponding to 1 × 107 CFU mL^−1^). The bacterial suspension (1 × 10^7^ CFU mL^−1^, 1 mL) was incubated in a PBS solution containing PMB-Se-NPs (32 µg mL^−1^, 1 mL) and subjected to LED lamp irradiation (450 nm, irradiance: 3 mW cm^−2^) or kept in the dark. The Se-MOF, Se-NPs, and PMB were administered at equipotent doses, with a control group that did not receive any sample treatment included. After incubation for 10 min, the mixture was washed with PBS. The bacteria were then resuspended in a PBS solution containing DCFH-DA (10 μM, 1 mL) for an additional 10 min. After the incubation, the fluorescence intensities following various sample treatments were detected using the FITC channel of the flow cytometer (NovoCyte Advanteon, Agilent, Santa Clara, CA, USA).

### 2.5. Antibacterial Assay

Antibacterial activity was investigated with propidium iodide (PI) staining. Initially, a *P. aeruginosa* bacterial suspension (1 × 10^7^ CFU mL^−1^, 1 mL) was treated with PMB-Se-NPs (32 μg mL^−1^, 1 mL). Subsequently, the resulting mixture was incubated for 10 min, either in a light-restricted environment or exposed to an LED light (450 nm). Simultaneously, the control samples comprising bacterial suspensions were treated with equivalent doses of Se-NPs or PMB. Then, the bacteria were redispersed in a PI solution (30 μM, 1 mL) to label dead bacteria. The fluorescence intensity in the PE fluorescence channel of the flow cytometer was monitored to investigate the antibacterial activity of different samples. 

### 2.6. Synergistic Antibacterial Analysis

The synergistic antibacterial activity of the PMB-Se-NPs was assessed by comparison with those of the Se-NPs with PMB added at an equivalent dose. PI staining was utilized to evaluate bacterial viability and determine the degree of bacterial cell death, which could be quantified through flow cytometry. The bacterial suspension (1 × 10^7^ CFU mL^−1^, 1 mL) was first incubated with varying concentrations of PMB-Se-NPs in a PBS solution (16, 32, and 64 μg mL^−1^) for 10 min under light conditions (wavelength: 450 nm, irradiance: 3 mW cm^−2^). The experimental concentration of the nanoparticles was selected based on the minimum inhibitory concentration (MIC) results. The determination of the minimum inhibitory concentration (MIC) was conducted following the guidelines outlined in the CLSI Performance Standards for Antimicrobial Susceptibility Testing M100, 32nd edition. Untreated bacteria were used as the control. Subsequently, the suspension was centrifuged and the bacteria were resuspended in a PI solution (30 μM, 1 mL) to label dead bacteria. The fluorescence intensity of the PE channel was then measured using flow cytometry.

### 2.7. Targeted Bacterial Analysis

The bacterial targeting ability of the PMB-Se-NPs was detected using the FITC channel of the flow cytometer. The presence of a benzoselenodiazole fluorescent group in the MOF nanoparticles enabled their quantitative detection through flow cytometry when they were bound to bacteria. The *P. aeruginosa* bacterial suspension (1 × 10^7^ CFU mL^−1^, 1 mL) was incubated with PMB (1 mg mL^−1^, 1 μL) or without PMB pretreatment for 5 min. After incubation, the suspension was centrifuged to remove any free PMB, and the bacteria were then resuspended in a solution of PMB-Se-NPs in PBS (16 μg mL^−1^, 1 mL). The mixture was incubated for an additional 5 min to allow for the binding of the nanoparticles to the bacteria. Bacteria without any added samples were included as a control. Concurrently, Staphylococcus aureus (*S. aureus*, ATCC 25923) was employed as a representative of Gram-positive bacteria to undergo an identical treatment. Following incubation, the samples were analyzed using flow cytometry to quantify the binding of bacteria to the nanoparticles. 

### 2.8. Crystal Violet Staining of Bacteria Biofilms

Sterile round coverslips were placed in 24-well plates as a substrate for biofilm formation. Bacterial suspensions (1 × 10^8^ CFU mL^−1^, 2 mL) were added to a 24-well plate, which was then incubated at 37 °C for 24 h. After that, the medium was removed and the obtained biofilm was incubated in 1 mL of media containing PBS (control), PMB (3 μg mL^−1^), Se-NPs (61 μg mL^−1^), or PMB-Se-NPs (64 μg mL^−1^). After 3, 6, or 12 h of incubation, an LED light of 450 nm was used to irradiate the biofilms for 30 min, corresponding to an optical power density of 3 mW cm^−2^. Following irradiation, the biofilms were washed thrice with PBS. Crystal violet (CV) staining was used for quantifying and visualizing the remaining biomass in the biofilms. The biofilms were first fixed with a methanol solution and subsequently stained with a CV solution at a concentration of 0.1% *w*/*v*. The residual biofilms were allowed to bind with the staining solution for 30 min. Following staining with CV, the biofilms were washed thrice with PBS to eliminate any unbound CV stain before being dissolved in a 95% *v*/*v* ethanol solution. The biomass of the biofilm was ultimately determined by measuring the OD value at 590 nm.

### 2.9. Live/Dead Staining Analysis

The biofilms of *P. aeruginosa* were obtained by static incubation at 37 °C using an LB medium in 35 mm glass-bottom dishes. After the bacterial suspension (1 × 10^8^ CFU mL^−1^, 2 mL) was incubated in the dishes for 24 h, the medium was removed and the biofilms were washed with PBS. The biofilms were incubated in 1 mL of media containing PMB (3 μg mL^−1^), Se-NPs (61 μg mL^−1^), or PMB-Se-NPs (64 μg mL^−1^) for 12 h. After the LED light source (450 nm, 3 mW cm^−2^) was used to irradiate the biofilms for 30 min, the biofilms were gently washed with PBS. Finally, the biofilms were stained with a Live/Dead Bacterial Viability Kit for 20 min and washed thrice with PBS. The biofilms after different treatments were observed via confocal laser scanning microscopy (A1, Nikon, Tokyo, Japan).

## 3. Results and Discussion

A Se-MOF was synthesized in DMF using acetic acid as a regulator. As shown in Figure 2A, it was observed with scanning electron microscopy (SEM) that the Se-MOF had an octahedral morphology. The average diameter of the Se-MOF was approximately 1.2 µm. To prepare small MOF-based nanoparticles, Se-NPs were synthesized in the presence of benzoic acid as a regulator and Tween 80 as a surfactant. In comparison to acetic acid, benzoic acid is more acidic, which helped to further inhibit MOF growth, resulting in the production of smaller-sized Se-NPs. Additionally, the presence of Tween 80 effectively prevented particle agglomeration during the growth process. According to the SEM image in Figure 2B, the average particle size of the Se-NPs was calculated from the statistical results and was around 68 nm with a standard deviation of 10 nm. Moreover, the morphology of the Se-NPs was further evaluated using transmission electron microscopy (TEM). As shown in Figure 2C,D, the Se-NPs were irregular in shape. Notably, the Se-NPs exhibited excellent dimensional stability even after being incubated in an LB culture medium for up to 24 h, with no signs of degradation observed. The TEM results are provided in Appendix A.

Powder X-ray diffraction (XRD) measurements were conducted to analyze the crystalline structures of the Se-MOF and Se-NPs. As depicted in Figure 2E, the XRD pattern of the Se-MOF exhibited three intense peaks at 2θ = 4.69°, 5.39°, and 9.38°, indicating its highly crystalline nature, which is consistent with the simulated UiO-68 framework. For the Se-NPs, no distinct 2θ peaks were observed, indicating that the Se-NPs were amorphous. Next, the ^1^O_2_ generation efficiencies of the Se-MOF and Se-NPs were evaluated using electron spin resonance (ESR). Two samples were prepared with dispersion in PBS (pH 7.4) at the same concentration of 500 μg mL^−1^. Then, the ROS-trapping agent 2,2,6,6-tetramethylpiperidine (TEMP) was added to the sample solutions to detect singlet oxygen generation before and after irradiation using a blue LED lamp with a wavelength of 450 nm. The excitation spectrum and characteristics of the LED lamp can be found in Appendix A. As shown in Figure 2F, there was no obvious signal for the Se-MOF or the Se-NPs before the light irradiation. However, a typical triplet signal characteristic from TEMP-1-oxyl appeared after the light irradiation (irradiance: 3 mW cm^−2^) for 5 min, demonstrating the generation of ^1^O_2_ from the Se-MOF and Se-NPs. Interestingly, the Se-NPs exhibited a considerably higher relative ^1^O_2_ yield under light irradiation compared to the Se-MOF. This result can be attributed to the smaller size of the nanoparticles with a larger specific surface area, which is more conducive to the diffusion of ^1^O_2_ from Se-NPs, as has been previously reported by Deng et al. [44].

Dynamic light scattering (DLS) and zeta potential measurements were performed to investigate the hydrodynamic size and zeta potential of the Se-NPs in PBS (pH 7.4) before and after conjugation with PMB. As shown in Figure 3A, the Se-NPs presented limited size distribution, with an average hydrodynamic diameter of 68.0 nm (PDI, 0.215), when they were dispersed in PBS at pH 7.4. Compared with the Se-NPs, the PMB-Se-NPs had larger hydrodynamic diameters (295 nm) and a broader size distribution (PDI, 0.384) in the PBS (pH 7.4). The larger diameters of the PMB-Se-NPs compared to the Se-NPs could be attributed to the hydration sphere and the outer PMB layer. In addition, the particle sizes of the PMB-Se-NPs were characterized via TEM. The result indicated that the diameters of the PMB-Se-NPs exhibited minimal changes in comparison with the Se-NPs (Appendix A). The zeta potential results presented in Figure 3B indicate that the Se-NPs had a positive zeta potential (+18.5 mV) when they were dispersed in the PBS buffer at pH 7.4. The observed result can be attributed to the introduction of the benzoic acid, a modulator with a high pKa. This addition resulted in an increased number of deprotonated modulator molecules, which effectively occupied a significant proportion of the available carboxylate sites. As a result, the surfaces of the nanoparticles acquired a positive charge, with an excess of Zr^4+^ ions compared to the negatively charged carboxylate groups. However, the surface charge increased to +24.0 mV after the PMB was modified onto the surface of the Se-NPs. This increase in surface charge can be attributed to the modification of the PMB molecules with multiple amino groups on the surfaces of the nanoparticles. The UV-vis spectra of the Se-NPs and PMB-Se-NPs were measured after they were dispersed in the PBS. As shown in Figure 3C, two absorption bands, at 205 and 275 nm, were observed for the Se-NPs. Modification of the Se-NPs with PMB led to a red-shift from 275 to 287 nm, indicating that the surface nature of the nanoparticles was changed and PMB was conjugated on the particle surfaces.

To investigate the binding between the PMB and the Se-NPs, Fourier transform infrared (FT-IR) spectroscopy was also performed, as depicted in Figure 3D. For the Se-NPs, a distinctive absorbance at 1600 cm^−1^ was observed; this was caused by the stretching vibration of the carbonyl C=O group. However, this peak disappeared after the Se-NPs were conjugated with the PMB, and the FT-IR spectra of the PMB-Se-NPs exhibited two new characteristic peaks at 1741 cm^−1^ and 1063 cm^−1^, which confirmed the formation of amide bonds between the PMB and the Se-NPs in the PMB-Se-NPs. All these results confirm that the PMB was covalently linked via amide bonds onto the surfaces of the Se-NPs. Next, the amount of PMB immobilized on the PMB-Se-NPs was determined using the classical ninhydrin hydrate method [46]. The amino groups in the PMB could react with hydrated ninhydrin in an aqueous solution to form a blue–purple substance that would have a maximum absorption wavelength of 570 nm. UV-vis spectroscopy was utilized to measure the absorbance at 570 nm, which was then used to construct a standard curve of absorbance vs. PMB concentration (Appendix A). The PMB content in the PMB-Se-NPs was estimated to be 4.7 wt%.

To evaluate the photodynamic efficacy of the PMB-Se-NPs, the ^1^O_2_ generation inside the bacteria was monitored using a DCFH-DA ROS Assay Kit. Non-fluorescent DCFA-DA could permeate live bacteria and be oxidized by ^1^O_2_ to produce dichlorofluorescein (DCF) compounds that would emit strong green fluorescence. The DCF fluorescence in the bacteria could be monitored using the FITC fluorescence channel of the flow cytometry. Without light irradiation, negligible fluorescence was detected from the bacteria treated with both the Se-MOF and the Se-NPs (Figure 4A). However, significant DCF fluorescence was detected inside the bacteria after treatment with two samples and light irradiation. Moreover, the Se-NPs had a higher level of ^1^O_2_ generation than the Se-MOF under light irradiation, which is in line with the ESR results (Figure 2F). In addition, the photodynamic activity of the PMB-Se-NPs was investigated and compared with the same concentrations of Se-NPs. The results showed that a much stronger fluorescence was found for the bacteria treated with the PMB-Se-NPs in contrast to those treated with the Se-NPs (Figure 4B). This outcome should have been because PMB can induce an increase in intramural ROSs in bacterial cells [47]. To investigate this issue, the ^1^O_2_ generation of PMB inside the bacteria was investigated. The bacteria incubated with the PMB-Se-NPs (32 µg mL^−1^, 1.0 mL) and the PMB (1.5 µg mL^−1^, 1.0 mL) were irradiated with a 450 nm LED lamp. Based on the percentage content of the PMB in the PMB-Se-NPs (4.7 wt%), the dosage of free PMB used for comparison was selected to be 1.0 mL of PBS that contained PMB (1.5 µg mL^−1^). The results showed that the PMB caused an increase in the ROS levels in the bacteria, which was unrelated to irradiation. Therefore, these results confirm that PMB-Se-NPs have synergistic antibacterial properties with photodynamic and antibiotic functions.

The synergistic antibacterial activity of the PMB-Se-NPs against *P. aeruginosa* was investigated with propidium iodide (PI) staining. PI is a red, fluorescent DNA-intercalating dye and can only penetrate damaged cell membranes. Bacteria with damaged cell membranes could bind with the PI and be monitored by the PE fluorescence channel of the flow cytometer. Both Se-NPs and PMB were used as control samples. The bacterial samples treated with the PMB showed that no significant difference in phototoxicity was observed with or without light exposure. This indicates that PMB is not photosensitive and the application of light does not significantly increase the percentage of bacterial death (Figure 5A). As shown in Figure 5B, no obvious cytotoxicity was observed for the Se-NPs in the dark. However, phototoxicity was observed clearly for the Se-NPs after light irradiation. This is attributed to the photodynamic activity of Se-NPs, which is activated under light conditions, leading to the generation of singlet oxygen that causes oxidative damage to the bacterial cell membrane. Compared with the Se-NPs, the PMB-Se-NPs exhibited higher phototoxicity against the bacteria (Figure 5C,D). The antibacterial activity of the PMB-Se-NPs was significantly enhanced, which could be explained due to the modification of the PMB on the surface of the Se-NPs. These results further confirm that the PMB-Se-NPs exhibited the synergistic therapeutic effects of antibiotic and photodynamic therapy.

To further confirm the synergistic bactericidal effect of PMB-Se-NPs, bacteria were incubated with an equal dosage of PMB-Se-NPs or a mixture of Se-NPs and PMB under light conditions. Bacterial apoptosis was assessed with PI staining, and the results are presented in Figure 6A. The flow cytometry results revealed that across three different concentrations, the PMB-Se-NP group exhibited superior bacteria-killing ability compared to the Se-NP and PMB mixture group. The results indicated that PMB-Se-NPs exhibit a synergistic effect in photodynamic/antibiotic therapy, surpassing the simple additive effects of Se-NPs and PMB alone. The superior bacteria-killing ability observed in the PMB-Se-NP group may be attributed to the bacteria-targeting capability of PMB-Se-NPs against Gram-negative *P. aeruginosa*.

Encouraged by the results of the bacterial apoptosis, the synergistic antibacterial mechanism of the PMB-Se-NPs was further investigated. The bactericidal mechanism of PMB has been extensively studied and reported [48,49]. PMB is known to bind to the negatively charged lipid A component of the outer membrane lipopolysaccharide (LPS) in Gram-negative bacteria. Based on the mechanism of PMB action, it is reasonable to hypothesize that PMB-Se-NPs would retain the LPS-targeting ability of PMB. The fluorescence emission spectroscopy results revealed that PMB-Se-NPs exhibited a fluorescence emission peak at 547 nm, primarily attributed to the inclusion of Se-NPs within the nanoparticle structure. Furthermore, the fluorescence emission properties of the Se-NPs were attributed to the incorporation of the benzoselenodiazole-containing ligand within the MOF structure (Appendix A). Therefore, upon the combination of the PMB-Se-NPs with the bacteria, the bacteria were effectively fluorescently labeled, enabling the quantification of the number of bacteria bound to PMB-Se-NPs through the FITC channel of the flow cytometer. Experiments were conducted to investigate the target binding of PMB-Se-NPs to two representative bacteria: *S. aureus* (Gram-positive) and *P. aeruginosa* (Gram-negative). The evaluation involved comparing the target binding of PMB-Se-NPs to these two bacteria in the presence and absence of PMB treatment. As illustrated in Figure 6B, the binding capacity of PMB-Se-NPs to *P. aeruginosa* was observed to be superior to that of *S. aureus* in the absence of PMB pretreatment. This difference can be attributed to the ability of PMB-Se-NPs to specifically target LPS present on the outer membranes of Gram-negative bacteria, facilitated by the presence of PMB on the nanoparticle surface. However, in the case of *P. aeruginosa*, a decrease in fluorescence intensity was observed following PMB pretreatment, indicating a reduced proportion of PMB-Se-NPs bound to the bacteria. This result suggests that the introduction of free PMB led to a decrease in the number of bacteria bound to PMB-Se-NPs, as some bacteria were already occupied by free PMB. Furthermore, it was observed that the presence or absence of PMB treatment had no significant impact on the binding of PMB-Se-NPs to *S. aureus*. The obtained results provide evidence of the targeted binding capability of PMB-Se-NPs to *P. aeruginosa*, which is attributed to the surface modification of PMB on the nanoparticles. Therefore, all these results revealed that the targeting of *P. aeruginosa* by PMB-Se-NPs increases the binding of nanoparticles and enhances phototoxicity in the outer membranes of Gram-negative bacteria under light irradiation, thus resulting in a synergistic antibacterial effect. 

Biofilm formation is known to protect bacteria from damage and promote bacterial invasion, resulting in severe infections. Therefore, antibiofilm activity is a crucial parameter for evaluating the efficacy of antibacterial agents. In this experiment, the crystal violet staining method was employed to investigate the conditions for biofilm formation. The results revealed that mature *P. aeruginosa* biofilms could be obtained by incubating 1 × 10^8^ CFU mL^−1^ of bacteria for 24 h (Appendix A). To evaluate the effectiveness of PMB-Se-NPs against biofilms, a biofilm model of *P. aeruginosa* was constructed, and the remaining biofilm was statistically analyzed using a CV staining method. After being incubated with different samples, the biofilms were exposed to an LED light at 450 nm (irradiance: 3 mW cm^−2^) for 30 min. The biofilm elimination ability of the PMB-Se-NPs was investigated with CV staining. Both PMB and Se-NPs were also evaluated as control samples. After 3 h of incubation, no biofilm elimination effects under light irradiation were observed for the three samples (Figure 7A). However, the biofilm biomass removal ability of the PMB-Se-NPs improved with the increase in incubation time. After 6 and 12 h of incubation, the biofilm elimination efficiency of the PMB-Se-NPs was significantly higher than those of the Se-NPs and PMB (Figure 7B). These results could be attributed to the synergistic effect of photodynamic therapy and antibiotics. Upon exposure to light, PMB-Se-NPs can produce a substantial quantity of singlet oxygen, resulting in oxidative damage of bacterial cells and biomolecules within a biofilm. Simultaneously, this process can effectively disturb the intricate network architecture of the biofilm and expose the dormant bacteria residing within it. Moreover, owing to their small size, PMB-Se-NPs can easily penetrate a biofilm, facilitating efficient delivery of PMB within the biofilm and augmenting its bioavailability. As a result, PMB-Se-NPs exhibit remarkable antibiofilm activity under light conditions, surpassing the efficacy of Se-NPs or PMB alone.

Additionally, live/dead staining was used to evaluate the biofilm elimination ability of PMB-Se-NPs. After being incubated with different samples for 12 h, the biofilms were irradiated with light for 30 min and then co-stained with the SYTO 9 dye and propidium iodide (PI) to distinguish between live (green) and dead (red) bacteria cells. As shown in Figure 7C, a low level of red fluorescence was detected in the bacteria treated with PMB, indicating a low efficiency of PMB in eliminating biofilms. This outcome can be attributed to the poor penetration of PMB and the resistance to PMB of bacteria living in biofilm communities. In the Se-NP treatment group, the fraction of the green fluorescence signal representing live bacteria within the biofilm was significantly diminished. These results suggest that the photodynamic activity of Se-NPs can be activated under light conditions, leading to the generation of singlet oxygen capable of disrupting the biofilm and significantly loosening its structure. Notably, almost no live bacteria were found after treatment with PMB-Se-NPs. The above results are consistent with the CV staining assay. All these results demonstrated that PMB-Se-NPs exhibit synergistic antibiotic and photodynamic effects in eradicating bacteria biofilms.

## 4. Conclusions

In summary, we synthesized two benzoselenadiazole-doped MOFs and evaluated their efficiency of ^1^O_2_ generation under irradiation with a 450 nm lamp (irradiance: 3 mW cm^−2^). Upon light exposure, the ^1^O_2_ generation efficacy of the Se-NPs was higher than that of the Se-MOF, possibly because small-sized Se-NPs are conducive to diffusion of ^1^O_2_. After PMB was coupled to the surface of Se-NPs through amidation, the PMB-Se-NPs exhibited superior bactericidal ability compared to free PMB and Se-NPs. Notably, the antibacterial mechanism of the PMB-Se-NPs primarily involved targeting Gram-negative bacteria through surface modification of PMB. This distinctive characteristic contributed to an antibacterial effect that surpassed the mere combination of Se-NPs and PMB. Crucially, the most remarkable feature of PMB-Se-NPs lies in their ability to effectively eradicate bacterial biofilms at relatively low antibiotic doses and light intensities thanks to the synergistic effect of antibiotic and photodynamic therapy. In short, synergistic antibiotic and photodynamic therapy with PMB-Se-NPs provides a promising strategy for fighting against biofilm infections.

## Figures and Tables

**Figure 1 pharmaceutics-15-01826-f001:**
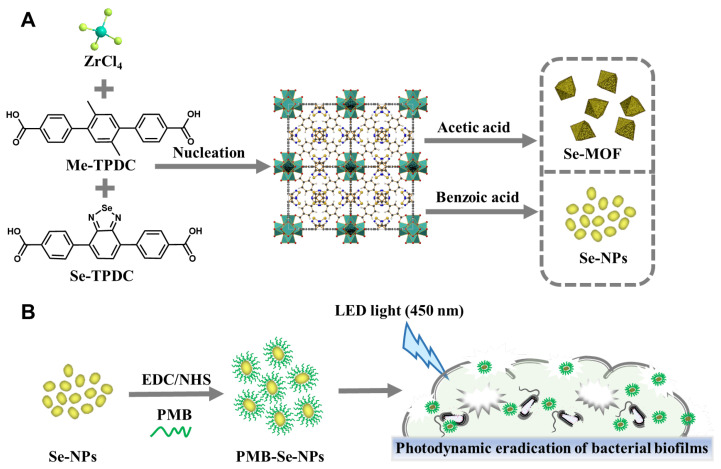
Schematic illustrations (**A**) Synthetic representation of Se-MOF and Se-NPs. (**B**) Schematic illustration of PMB-Se-NPs for eradicating bacterial biofilms through synergistic photodynamic and antibiotic therapy.

**Figure 2 pharmaceutics-15-01826-f002:**
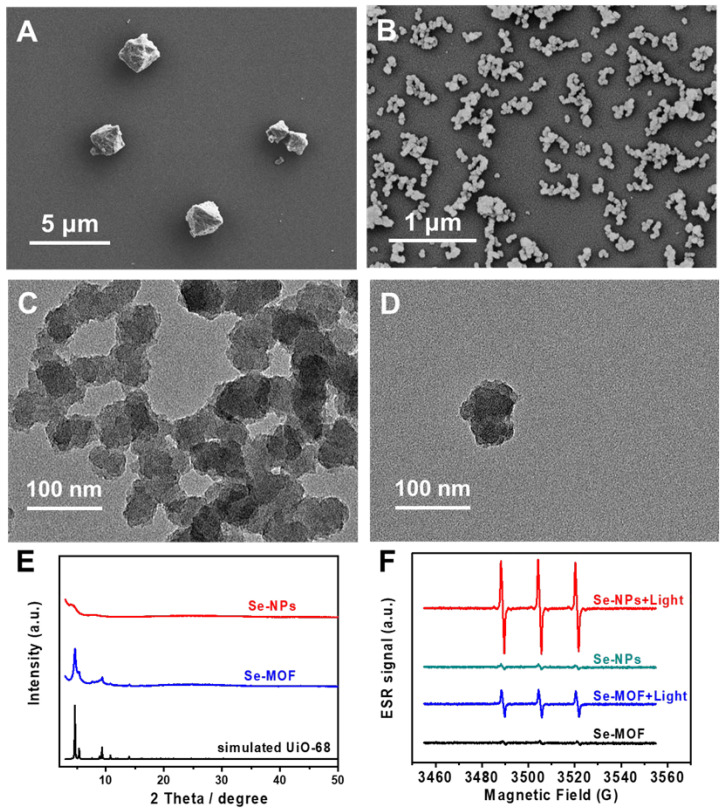
Characterizations of Se-MOF and Se-NPs. SEM image of as-synthesized (**A**) Se-MOF and (**B**) Se-NPs. (**C**,**D**) TEM images of Se-NPs. (**E**) XRD profiles of Se-MOF and Se-NPs. (**F**) ESR spectra of Se-MOF and Se-NPs in a PBS solution before and after illumination.

**Figure 3 pharmaceutics-15-01826-f003:**
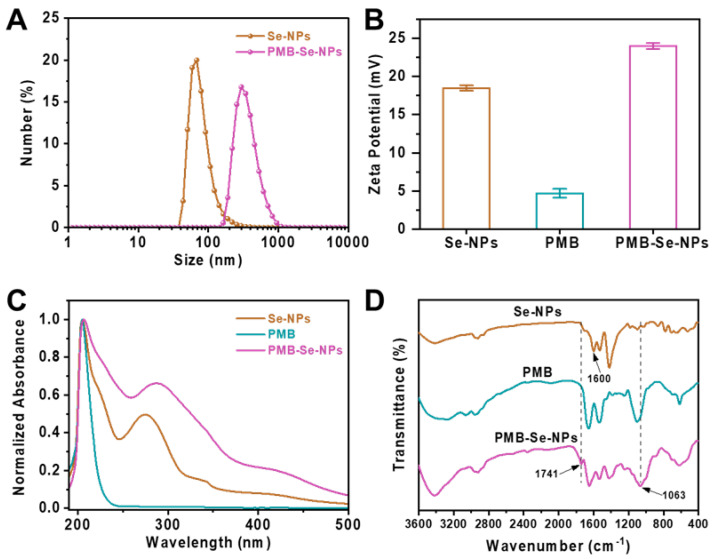
Characterization of PMB-Se-NPs. (**A**) The hydrodynamic diameters of Se-NPs and PMB-Se-NPs. (**B**) The zeta potentials of Se-NPs, PMB, and PMB-Se-NPs. (**C**) UV-vis absorptions and (**D**) FT-IR spectra of Se-NPs, PMB, and PMB-Se-NPs.

**Figure 4 pharmaceutics-15-01826-f004:**
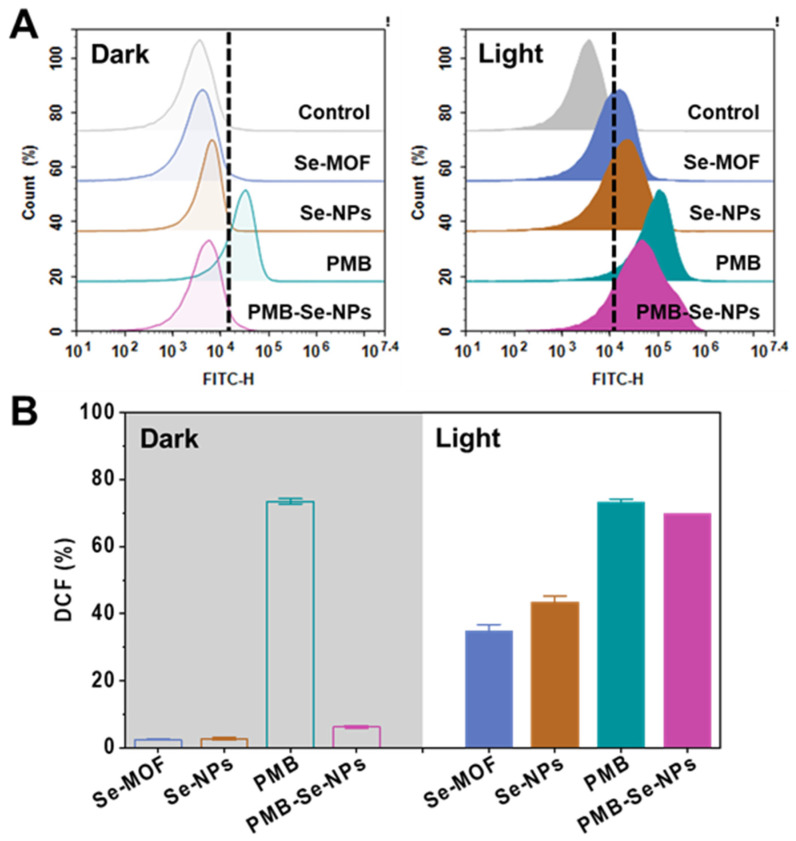
Intracellular ROS assay of bacterial cells. (**A**) Flow cytometry analyses showing that intracellular ^1^O_2_ was detected with DCFH-DA staining after *P. aeruginosa* bacteria were exposed to various samples with or without 450 nm light irradiation (irradiance: 3 mW cm^−2^). (**B**) Average DCF fluorescence intensity obtained from the bacteria in (**A**). (FITC-H: fluorescence intensity of DCF; Results are presented as means ± S.D; *n* = 3).

**Figure 5 pharmaceutics-15-01826-f005:**
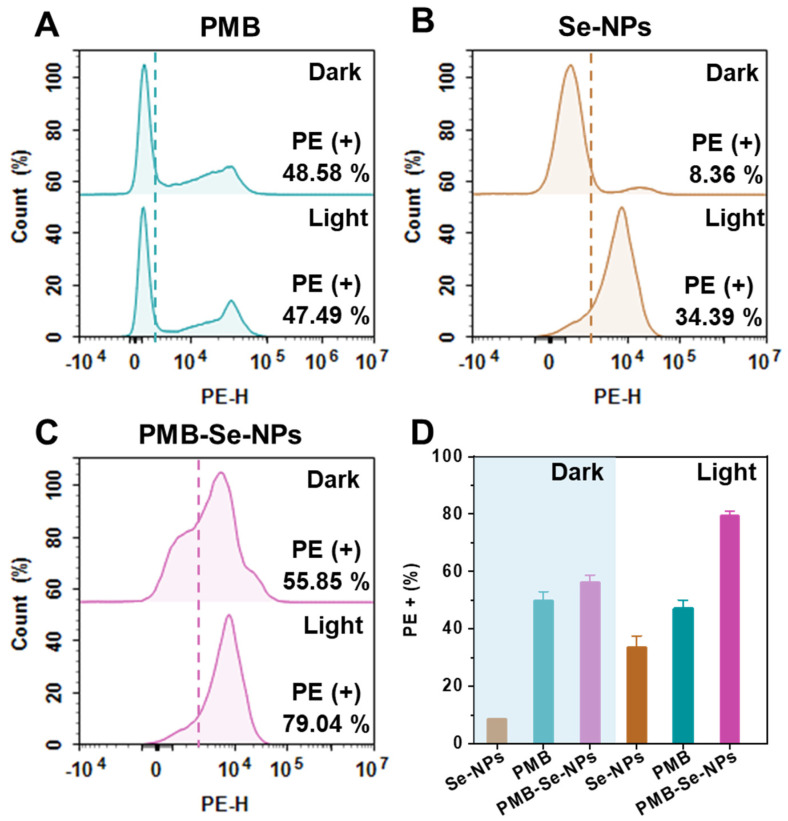
Bacterial apoptosis assay. Flow cytometry analysis was performed to detect the presence of apoptotic bacteria in *P. aeruginosa* samples that were treated with or without 450 nm light (irradiance: 3 mW cm^−2^). The bacterial samples were treated with (**A**) PMB, (**B**) Se-NPs, and (**C**) PMB-Se-NPs. (**D**) Average PI fluorescence intensity obtained from the apoptotic bacteria in (**A**–**C**). (PE-H: fluorescence intensity of PI; results are presented as means ± S.D; *n* = 3).

**Figure 6 pharmaceutics-15-01826-f006:**
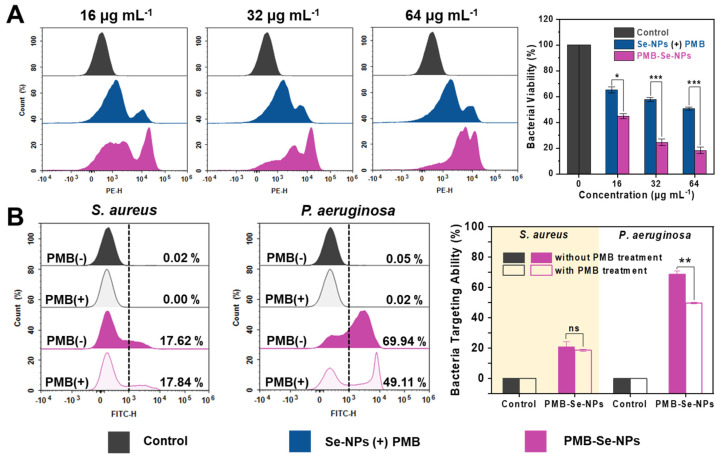
Synergistic antibacterial activity and mechanism analysis of PMB-Se-NPs. (**A**) Flow cytometry analysis of PI−stained bacteria treated with Se-NPs in combination with a PMB physical mixture or PMB-Se-NPs at concentrations of 16, 32, and 64 µg mL^−1^, following a 10 min light exposure incubation (450 nm, irradiance: 3 mW cm^−2^). (Results are presented as means ± S.D; *n* = 3. * *p* < 0.05 and *** *p* < 0.001). (**B**) Flow cytometric analysis was performed to evaluate the bacteria-targeting ability of PMB-Se-NPs toward *S. aureus* (Gram−positive) and *P. aeruginosa* (Gram−negative) strains, with and without PMB treatment. (Results are presented as means ± S.D, *n* = 3. ns = not significant; ** *p* < 0.01).

**Figure 7 pharmaceutics-15-01826-f007:**
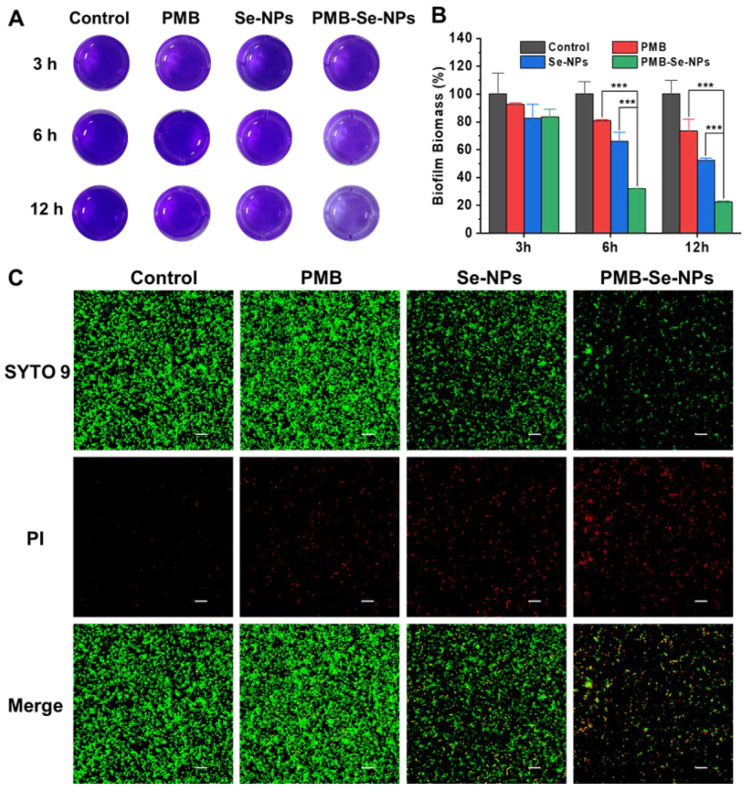
*P. aeruginosa* biofilm elimination. (**A**) Representative images of CV−stained biofilms after incubation for different times (3, 6, and 12 h), followed by irradiation with light at 450 nm for 30 min (irradiance: 3 mW cm^−2^). PBS was used as a control. (**B**) The biomass was quantified by measuring the OD value of the CV−stained biofilm in (**A**). (Results are presented as means ± S.D; *n* = 3; *** *p* < 0.001). (**C**) Representative live/dead staining images of *P. aeruginosa* biofilms after being incubated with different samples for 12 h and then irradiated for 30 min. Live bacteria were stained with green dye (SYTO9), and red dye (PI) was used to stain dead bacteria. Scale bar: 20 µm.

## Data Availability

Not applicable.

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
