# Peer review of "Synergistic Photodynamic/Antibiotic Therapy with Photosensitive MOF-Based Nanoparticles to Eradicate Bacterial Biofilms"

_pharmaceutics, 2023, doi:10.3390/pharmaceutics15071826_

Round 1

Reviewer 1 Report (New Reviewer)

The manuscript entitled “Synergistic Photodynamic/Antibiotic Therapy with Photosensi-2 tive MOF-based Nanoparticles to Eradicate Bacterial Biofilms » is submitted by L. Luan and Q. Zhang for publication in Pharmaceutics. In this probably revised version of the manuscript, the authors report their synthesis of two benzoselenadiazole-doped MOFs (Se-NPs and related PMB-Se-NPs decorated ones, through amidation), and evaluation of their efficiency in singlet O2 generation under a 450 nm wavelength irradiation. It is claimed that PMB-Se-NPs exhibited superior bactericidal ability compared to free PMB antibiotic and Se-NPs. Notably, the antibacterial mechanism of PMB-Se-NPs involves targeting Gram-negative bacteria through the surface modification of PMB. The quite high efficiency of PMB-Se-NPs should lie in their synergistic effect of antibiotic and photodynamic therapy (PDT); which could represent a promising strategy for fighting against biofilm infections.

Remarks and comments:

- Figure 3 C reveals light scattering for the NPs; do the authors agree ?

- Figure S4 corresponding to fluorescence emission spectra (λex = 390 nm) of PMB-Se-NPs and PMB dispersed in PBS buffer is not clear ; what is the very strong signal for PMB-Se-NPs  at higher energy?

In conclusion, this manuscript is quite well written, of interest in the domain and could deserve for publication in Pharmaceutics, but I need comment and answer to the asked question first; it is for me crucial as far as optical characterizations of the NPs are concerned.

Author Response

Reviewer 2 Report (New Reviewer)

This article by Du et al. describes the synthesis of photoactive metal-organic-frameworks (MOF)-derived nanoparticles (NPs) intended for antibacterial applications, with potential activity on biofilms. The NPs are compared to their MOF (larger) analogs in terms of ROS (singlet oxygen in this case) production capacity under visible light irradiation and further studied for their capacity to induce bacterial apoptosis and biofilm elimination on Pseudomonas aeruginosa.

The article is well written and easy to read. The results are clearly presented and support the conclusions. The experimental details are properly reported. I believe this is a quality article worthy of publication in Pharmaceutics after the following concerns are addressed:

11-      The 4-fold increase in diameter of the NPs upon PMB conjugation is intriguing. It seems rather difficult to accept the authors’ hypothesis that such a size difference is due to “hydration sphere” and “PMB layer”, especially considering that PMB represents less than 5 wt% of the final object. Is there no risk of NP aggregation with PMB serving as a reticulation agent? Each PMB molecule has several (5) free amines and could thus potentially link multiple NPs to one another…

22-      How do the authors explain that the zeta potential of Se-NPs is positive before PMB conjugation? One would expect the surface carboxylates to convey a negative surface charge. The increase of the zeta potential to a slightly higher value upon PMB-conjugation seems rather limited. PMB itself bears multiple primary amines (ammoniums in PBS) and if the NPs are indeed covered with PMB (to the point that they become so much bigger), shouldn’t they have a very positive zeta potential?

33-      The authors should investigate the morphology of Se-NPs by TEM after PMB conjugation.

44-      The final diameter of PMB-decorated NPs is around 300 nm. How is this size compatible with the diffusion of the objects through biofilms?

55-      It would have been interesting to have the Se-MOFs included in the different antibacterial assays, in order to better evidence the advantage of working with smaller NPs.

Author Response

Reviewer 3 Report (New Reviewer)

The current manuscript (pharmaceutics-2448626) presented their investigation on the preparation of a photosensitive metal-organic framework (MOF) nanoparticle with potential anti-bio effect. Their data indicated that, under 450 nm lamp exposure, the nanoparticles showed a higher oxygen generation efficacy comparing with the particle with micron size. From this, an effective antibacterial property was observed with bacteria P. aeruginosa. Overall, this work is interesting to clarify the mechanism of anti-bio effects with artificial nanomaterials. However, some points should be clear before the accept of current work.

1. About the materials, based on the Figure 1B, 1C, the aggregation of nanoparticles was showed by both SEM and TEM, which is general happened. However, considering about the Figure 2A and 2B, is the PMB surface grafting leading to more serious aggregation of nanoparticles, as the size was increased greatly. Please discuss this point.

2. As showed in Figure 4, the anti-bio property was show only under the light exposure. Please re-consider the potential application of prepared nanoparticles and discuss this point.

3. Appropriate controls should add for anti-bacterial testing, for example only add the PMB or only add the nanoparticles. In addition, could you please explain the concentrations selected (16, 32, and 64μg mL-1) for antibacterial assay. If it is possible, please consider using standard operation, for example ISO methods.

Round 2

Reviewer 1 Report (New Reviewer)

Concerning the new version of this article, the authors tried to answer the comments; I am still concerned by a question: Fig S5 (new version) describes emission properties of studied systems ; excitation occurs at 390 nm and emission is around 550 nm; please what is the signal in between ? Regards

Author Response

Reviewer 3 Report (New Reviewer)

The current manuscript has been improved according to the suggestions of reviewers. The answers to my questions are acceptable. This work is interesting and important to clarify the mechanism of anti-bio effects with artificial nanomaterials.

Author Response

Thanks for your recommendation of publication.

Round 3

Reviewer 1 Report (New Reviewer)

With the comments of the authors and even if quality of the photophysical comments still could have been improved, i think this manuscript can now be published in Pharmaceutics.

This manuscript is a resubmission of an earlier submission. The following is a list of the peer review reports and author responses from that submission.

Round 1

Reviewer 1 Report

Dear Editor,

The paper reports synthesis of nanoparticles from pre-synthesized metal organic frameworks.  A series of tests was performed to show whether they are eligible to be proposed as possible agents in photodynamic therapy against Pseudomonas aeruginosa biofilms. The paper is well-written. Before it gets online, there are some points should be addressed.

1- Full name of Se must be written.

2- Stability of MOFs should be discussed- there should be at least a theoretical explanations for their stability.

3- There should be some deep mechanistic discussions for the formation of nanoparticles.

Kind regards,

Reviewer 2 Report

The manuscript pharmaceutics-2359205 entitled " Synergistic Photodynamic/Antibiotic Therapy with Photosensitive MOF-based Nanoparticles to Eradicate Bacterial Biofilms” submitted by Q. Zhang and co-workers describes the preparation of nanoscaled benzoselenodiazole-doped MOFs (Se-NP) . The MOF nanoparticles surface were modified by polymyxin B to improve the ability of this nanoparticles to eradicate bacterial biofilms trough the combine action of Polymyxin B and the photodynamic action.

The work in a follow-up of the previous work performed by the authors in which used Se_MOF with an average of 1.2 micrometer and demosntrate do not so effective to erradiacae biofilms. In this work the authors not only reduced the size of the Se-MOF to ~70 nm but also add on the MOF surface the polymyxin B, an antibiotic drug.

The authors claimed a synergic effect but the results don't support this statement. Moreover many of the tecniques and experimental details are not clear and we are not able to reproduce them. So,  the manuscript need to be significantly improve to be consider for publication on the Pharmaceutics journal.

Main concerns:

a. The title is misleading. No synergistic effect was obtained. At least the authors should clearly demonstrate this fact. The authors probably obtained an additive effect: antibiotic + PDT.

b. The material and methods of the section are very incomplete. Equipment, purity of reagents and where they were purchased, as well as all experimental assays performed (DLS, SEM, TEM, XRD, 1O2, FTIR, etc.), should be described.

c. The way how the materials were dry should be added in section 2.1 and 2.2

d. The number of bacteria present at OD600 nm=0.1 must be added. It is not clear how the PDT experiments were performed, which volume was used and what was the bacterial concentration.

e. There is no information about the limit of detection of the quantification methods used to quantify bacterial reduction.

f. There is no information about the maturation of the bacterial biofilm. Did the authors carry out experiments to verify biofilm formation?

g. In the assay 2.2. the OD had been changed. Is there any explanation for this fact? No information again about the volume used.

h. Please add information about how the light irradiance was measured,the emission spectrum of the LED used and its characteristics.

i. In lines 204-205, the authors stated “Interestingly, Se-NPs exhibited a considerably higher relative quantum yield of 1O2 generation under light irradiation compared to Se-MOF”. Considering that the quantum yield is based in the number of photons absorbed by each material, did the authors take this aspect into account? The description of how the 1O2 tests were performed must be clarified.

j. The manuscript presents in the results and discussion section several parts that are not results or discussion, but experimental description. Please remove these sentences from this section.

k. How did the authors confirm the synergistic antibacterial properties?

I. No information about the meaning of FITC-H or PE-H in figures 4 and 5.

m. No information related to the number of independent experiments nor the repetitions. Statistical analysis must be described.

 Considering the number of flaws and the missing information which do not allow to confirm all the data present, I regret not recommending this manuscript for publication.

Reviewer 3 Report

Dear Authors,

Thank you for the interesting manuscript.

I appreciate its high quality, including the data presented and the figures clarity. I only recommend to take into account some comments:

1. The "Results and Discussion" section has to be supplemented with a generalizing table containing some comparison with an earlier achieved results from the literature sources. 

2. The section "Conclusions" should be expanded considering the additional information given in the supposed generalizing table in the previous section. It is needed to emphasize the novelty of the obtained results for the readers.

3. Figure captions should have a generalizing part, for example "Figure 1. Results of analysis of the structure of samples: (a) Electron microscopy data; (b) XRF results...".

4. Line 119, "105 °C for 24 h..". The extra dot at the end of a sentence.

5. Line 178, "Se-MOF was synthesized according to our previously reported method". The sentence seems redundant to me because it repeats the information given in the Section 2.1.

6. Line 189, "PXRD profiles of Se-MOF". There is no "PXRD" abbreviation introduction in the text. I think "XRD" should be used through out the whole text.

7. Lines 363-366. The information on the financial support duplicates the one given in the Section "Funding".